# High Consistency Silicone Rubber Foams

**DOI:** 10.3390/polym16091181

**Published:** 2024-04-23

**Authors:** Timo Hofmann, Ralf-Urs Giesen, Hans-Peter Heim

**Affiliations:** Polymer Engineering, Institute of Material Engineering, University of Kassel, 34125 Kassel, Germany; giesen@uni-kassel.de (R.-U.G.); heim@uni-kassel.de (H.-P.H.)

**Keywords:** silicone foams, extrusion, microspheres, water, chemical blowing agent, peroxid

## Abstract

Silicone elastomers are high-performance plastics. In the extrusion process, only high-consistency silicone rubbers were used. In order to reduce the cost and weight, silicone rubbers can be foamed during processing. In this study, high-consistency silicone rubber is processed with different physical and chemical blowing agents. The resulting reaction kinetics, as well as the mechanical and morphological properties, had been investigated and compared with each other. This showed that the chemical blowing agent significantly influenced the crosslinking reaction compared to the microspheres and the water/silica mixture tested, but it also achieved the lowest density compared to the physical blowing agents. When evaluating the foam morphology, it became clear that the largest number of pores was achieved with the microspheres and the largest pores when using the water/silica mixture. Furthermore, it has been shown that the different mechanisms of action of the blowing agents have a major influence on the mechanical properties, such as the micro shore hardness and the foam morphology.

## 1. Introduction

In the field of plastics technology, the application and processing of silicone rubber is becoming more and more important because it is easy to process. Products made of silicone rubber have high-temperature stability and good resistance to chemicals. Responsible for these properties is the molecular structure of silicone, which consists of alternating silicon and oxygen bonds, resulting in special organic and inorganic properties [1,2].

For this reason, silicone elastomers are particularly suitable for automotive and medical applications [3].

In the field of hot-curing silicone rubbers, a distinction is also made between liquid silicone rubbers (LSR) and high-consistency silicone rubbers (HCR). The difference between the two types is that HCR has longer molecular chains and is usually crosslinked with organic peroxides during processing [1].

The organic peroxides used here are subdivided into vulcanization under pressure, as with a press, and without pressure, as with extrusion. In the case of extrusion, DCIBP (di-(2,4-dichlorobenzoyl) peroxide) and DMB (di-(4-methylbenzoyl) peroxide) are used. For processes under pressure, diacyl peroxides such as dibenzoyl peroxides or dialkyl peroxides such as dicumyl peroxide and 2,5- dimethyl-2,5di(tert-butyperoxy)hexanes can be used [4,5].

The characterization of the crosslinking process in the rubber sector is carried out as standard with the aid of a Rubber Process Analyzer [6,7]. Among other things, the elastic torque for characterizing the curing behavior of the compound and the Tc 90 value, which indicates the time until 90% crosslinking is achieved, are determined [4].

Typical manufacturing processes for silicone rubber are injection molding, compression molding, and extrusion, with extrusion accounting for the largest share in terms of volume [8].

During the extrusion of silicone rubber, as shown in Figure 1, the compounded material is fed through the extruder into the crosslinking unit, where the rubber is continuously vulcanized. The vulcanization methods that have become established for this process are vulcanization using hot air and infrared radiation [9,10].

In each of the fields of application described above, as with other plastics, it is possible to make the desired components lighter and less expensive by foaming the material with blowing agents [11].

In the case of rubber, various methods can be used to foam the material. One possibility, for example, is the injection of physical blowing agents such as carbon dioxide [12,13,14] or water [15] in the material before the vulcanization.

In the field of silicone rubbers, chemicals such as AJBN (azobisisobutyronitrile) or carbonates such as azodicarbonamide (ADCA) are usually used as chemical blowing agents, according to the current state of the art. These are often harmful to health, and by-products and degradation products from the reaction remain in the foam. In this case, the decomposition reaction of the blowing agent must be coordinated with the crosslinking reaction of the silicone rubber. The background to this is that both are thermally activated, which makes process control much more complex [16].

During the decomposition reaction of the blowing agent, azodicarbonamide, carbon monoxide, and nitrogen, as well as hydrazodicarbonamide and cyanic acid, are formed here due to the addition of heat [17]. However, these products are problematic when used with addition-curing systems because they are susceptible to catalyst poisons such as sulfur and nitrogen compounds [9].

The platinum complex of the catalyst forms stable complexes with the amine or organosulfur compounds, rendering the catalyst inactive [2].

Furthermore, it is known from studies on natural rubber that the addition of the chemical blowing agent influences the measured shear force when analyzing the crosslinking characteristics. The reason for this is that the degradation of the blowing agent and the formation of the blowing gases lead to the formation of micro-cavities, which reduce the shear force compared to a measurement without a blowing agent [18,19].

The crosslinking behavior of rubbers is usually measured using a Rubber Process Analyzer, which has a temperature-controlled test chamber with two biconical cavities. A shear load is applied to the sample by a sinusoidal oscillation of the lower plunger, and the torque behavior of the material is measured. The structure of this measuring unit is shown as an example in Figure 2 [4].

The alternative to chemical blowing agents is physical blowing agents. Here, gases such as carbon dioxide and nitrogen or liquids such as water can be injected into the extrusion process [15]. In the case of water, the blowing agent can also be incorporated into the rubber mixture using a carrier substance such as silica, which means that additional foaming equipment is not required [20].

When water is used, the foaming process is achieved by evaporation of the water, whereas when gases are used, foaming takes place as soon as the material leaves the mold due to the pressure drop that occurs [21].

The use of water as a blowing agent is currently researched for thermoplastic elastomers (TPE) [22], ethylene–propylene–diene rubber (EPDM) [23], acrylonitrile–butadiene rubber (NBR) [24], and liquid silicone rubber (LSR) [20].

Similar to the chemical blowing agents, previous studies have shown an influence on the crosslinking behavior, whereby the effects depend on the rubber used [24].

In the field of injection molding of LSR, it has been found that with water, an increase in the process temperatures results in a higher weight reduction and an increase in the Shore A hardness. In addition, the number of pores increases, but not the pore size [20].

Another alternative among the physical blowing agents is the use of thermoplastic microspheres, which are used, for example, in liquid silicone rubbers. These are mixed into the silicone rubber and react during vulcanization by supplying heat. The gas within the microspheres expands, which leads to a strong enlargement of the microspheres and, thus, to the foaming of the silicone rubber, as shown in Figure 3 [11].

In the area of silicone rubbers, however, the use of water and microspheres as blowing agents has only been researched in the field of liquid silicone rubbers [11,20,25], according to the current state of the art. There are still no published results in the field of high-consistency silicone rubbers.

## 2. Materials and Methods

The high-consistency silicone rubber (HCR) used was Elastosil R 401-40 S from Wacker Chemie AG (Munich, Germany), which was crosslinked with 1.5 phr of the peroxide Peroxan PMB (di-(4-methylbenzoyl)peroxide) from Pergan Hilfsstoffe für industrielle Prozesse GmbH (Bocholt, Germany). The Elastosil R 401-40 S has good flexibility and mechanical properties and is suitable for vibration damping, for example. The HCR type is a variant that is preferably crosslinked with peroxide. Unlike usual, no chlorine-containing system was used for the crosslinker, as these peroxides were to be replaced by chlorine-free alternatives due to the emissions of PCBs (polychlorinated biphenyls).

Both chemical and physical blowing agents were used. Pergan’s peroxane AZDN (2,2′-dimethyl-2,2′-azodipropiononitrile) was used as the chemical blowing agent. Thermoplastic expandable microspheres Unicell MS 140 DS from Tramaco (Tornesch, Germany) and a water/silica mixture in a ratio of 2:1 served as physical blowing agents. Therefore, a fumed hydrophil silica Aerosil 200 from Evonik Industries AG (Essen, Germany) with a specific surface area of 200 m^2^/g was used. The MS 140 DS microspheres used have an unexpanded particle size of approximately 20 µm and start to expand at a temperature of approximately 90 °C.

For the compounding of the HCR, a twin-screw internal mixer (CTM-25) from COLMEC SPA (Busto Arsizio, Italy) was used, which can mix 1.5 kg per batch. The batch consists of three components: rubber, a crosslinker, and a blowing agent. The mixing time for each was 20 min; see Figure 4. The mixing process is divided into three stages. First, the rubber is masticated, which aims to homogenize the material thermally and mechanically. In the second step, the blowing agent is incorporated into the rubber during filler incorporation. The aim here is to convert the two-phase system into a single-phase system. In the final step, the crosslinker is incorporated, and the single-phase system is homogenized. After the mixing process, the material was discharged in strips for further processing on the extruder.

The silicone extruder EEK 32.12 S—4/90 SIR from Rubicon Gummitechnik und Maschinenbau GmbH (Halle, Germany) was then used to produce 13 mm diameter strands at one speed and two temperatures, varying the diameter of the extrusion die depending on the blowing agent, as shown in Table 1. The temperature specified here is the temperature that is set on the infrared lamps. The energy for vulcanization is achieved by an infrared unit. The thermal energy required for vulcanization is achieved by the absorbed IR radiation, which excites the polar molecules to vibrate, generating heat [10].

In the characterization methods, the vulcanization behavior of the compounds was first examined as a function of the blowing agents used. The background to this investigation is that the blowing agents used may have an influence on the crosslinking reaction. As already described in the introduction, the formation of pores will result in a reduction in the maximum torque. For this purpose, a Rubber Process Analyzer from Montech (Buchen, Germany) D-RPA 3000 was used, which determined the vulcanization at 180 °C for 2 min with an angle of 1° and a frequency of 1.67 Hz. Approximately 5 g of the tested compounds were weighed per measurement and placed between two test films of polyethylene terephthalate with a thickness of 23 µm and then placed in the RPA. Two measurements were taken for each compound, from which the mean value was calculated.

Test specimens with a length of 10 mm and a diameter of 13 mm were cut from the extrudate itself in order to determine the mechanical properties. First, the micro shore A hardness was determined in the middle of the specimen with a Digi test from Bareiss (Oberdischingen, Germany). For each parameter, the micro shore hardness was determined for five different parameters.

Because of the different blowing agents, both open-pored and closed-pored foams were produced. For this reason, the density could not be determined using the Archimedean principle. Therefore, the height and diameter of the individual pucks were measured, and then their weight was determined in order to calculate the density. For each parameter, five test specimens were measured.

As a final parameter, the surface structure of the foams was examined with a VK-X3000 microscope from Keyence (Neu-Isenburg, Germany). For this purpose, 64 images were taken at 200× magnification over the cross-sectional area of the extrudate, and these were evaluated. The number and size of the pores over the diameter of the test specimen were analyzed.

## 3. Results

In the following section, the results of the vulcanization measurement, the mechanical and physical parameters, and the foam morphology, which were measured using the methods described above, are presented and then compared with comparable results from research.

### 3.1. Vulcanization Properties

When measuring the vulcanization behavior, a clear difference in the crosslinking between the individual blowing agents can be seen, as shown in Figure 5. When comparing the different blowing agents, there are clear differences in the torque curve when contrasting the chemical and physical blowing agents. With the chemical blowing agent, it can be seen that both the gradient of the torque curve and the maximum torque deviate significantly from those of the reference without and with the physical blowing agent. In contrast, the physical blowing agent only has a small influence on the vulcanization properties of the compound compared to the chemical blowing agents. Here, we see that the measurements only lead to a reduction in the maximum torque and not to a change in the crosslinking speed as with the chemical blowing agents.

### 3.2. Density

Figure 6 shows that the lowest density of approximately 0.4 g/cm^3^ can be achieved by the chemical blowing agent. With the physical blowing agents, the lowest value was achieved with the microspheres as a function of temperature, at 0.5 g/cm^3^ at 500 °C and 0.48 g/cm^3^ at 550 °C. In comparison, the water-based blowing agent only provides densities of 0.59 g/cm^3^ at 500 °C and 0.55 g/cm^3^ at 550 °C. However, with the water/silica mixture, it is clear that the temperature has the greatest influence on the density compared to the other blowing agents. Furthermore, it can be seen that, despite a similar mechanism of action, the AZDN and the water–silica mixture deliver clearly distinguishable results.

### 3.3. Hardness

In terms of hardness, it can be observed that the lowest values can be achieved when using the chemical blowing agent AZDN (between 5 and 8 micro shore A). In comparison, the used physical blowing agent microspheres exhibit the highest hardness (between 20 and 23 micro shore A) regardless of temperature, as shown in Figure 7. The water mix produces a micro shore A hardness between 7 and 13 for the lower temperature and 12 to 16 for the higher temperature. In addition, it can be seen that the samples foamed with the water/silica mixture show the greatest scatter in the measurements compared to AZDN and the microspheres.

### 3.4. Surface Structure

As described above, the surface structure was evaluated using the number of pores and pore size over the diameter of the round profile (referred to as the cross-sectional area) with the aid of the VK-X 3000 MultiFile Analyzer from Keyence. The evaluation of the pore numbers of the different blowing agents shows that there are large differences in the pore numbers between the blowing agents (see Figure 8). It is also clear that an increase in temperature leads to an increase in the number of pores for AZDN and microspheres as blowing agents. For AZDN, the number of pores determined increases from 163 to 288, and for microspheres, from 674 to 749. In contrast, no change occurs when using the water/silica mixture.

When looking at the cross-sectional area (Figure 9), it can be seen that there is an increase in the number of pores with a small cross-sectional area as a result of the temperature increase. In addition, it can be seen that the physical blowing agent water/silica mixture in the test produced the most large pores in relation to the number of pores. On the other hand, however, it is also clear that the microspheres produce the smallest and most pores, regardless of the temperature.

The different pore sizes of the blowing agents also become clear when looking at the microscopy images from the cross-section area, as shown in Figure 10, as an example for the chemical blowing agent AZDN and the physical blowing agent water/silica mixture. As can be seen in Figure 10, the AZDN and the water mix cause different pore formations during gas formation.

## 4. Discussion

Based on the results, here we discuss the reaction kinetics, pore distribution, and mechanical properties. With regard to the crosslinking kinetic, it can be confirmed that, as already described by Sitz [16], the crosslinking reaction of the peroxide and the chemical blowing agent influence each other. Similar to the findings in [18,19], there is a decrease in the elastic torque due to the formation of microvoids caused by the gas expansion. In the case of microspheres, the explanation could lie in the foaming mechanism. Since there is no free expansion of gases and the density reduction is achieved by increasing the size of hollow spheres, there may be no further cavities that lower the elastic torque. When comparing the results from the tests of [24] with EPDM and NBR with the HCR used here, there are clear differences in the influence on the crosslinking reaction. One possible reason for the difference between the results is the proportion and mixing ratio of water used. As higher proportions of water were mixed into the silica in Hopmann’s tests, this produced more gases, which can lead to the formation of cavities, as described in [18,19]. The effects of the different foaming mechanisms and the interaction with the crosslinking are also reflected in the resulting density. As shown in Figure 6, different densities occur with the same blowing agent content. One explanation for this could be the different expansion mechanisms of the respective blowing agents. In the case of microspheres, for example, the maximum expansion is determined by the thickness of the cell wall. In the case of the chemical blowing agent and the water/silica mixture, the differences can be explained by the different gas expansion quantities, as already indicated by the measurements on the Rubber Process Analyzer. By increasing the blowing agent content, the densities of the material could be further reduced in the case of the water/silica mixture, similar to the LSR, as described by Marl [20]. However, an increase in the expandable microspheres also led to a significant reduction in density, as investigated in [11].

The significant differences in the number of pores between the individual blowing agents depend on the different mechanisms of action, as described above. The number of pores in the silicone foamed with microspheres depends mainly on the number of microspheres previously incorporated. In addition, a change in temperature apparently influences the rate of expansion of the microspheres. In terms of density, this results in a higher final size of the microspheres, which is why the number of measured pores increases as the distance between the individual spheres decreases. With the gas expansion of AZDN and water, on the other hand, the number of pores is influenced by the amount of gas and the crosslinking speed, as shown in Figure 8. Despite a similar principle of action, these two blowing agents differ in terms of the number of pores and size.

As can be seen in Figure 10, pores of different sizes are formed with the same process parameters. One reason for this must lie in the different gas-development processes for foaming. While AZDN is a chemical decomposition, the gases in the water/silica mixture are produced by a phase transformation of the water. The different effects of the respective gases on the silicone can lead to differences in the pore size and number of pores. Based on the results of the foam structure, the differences in micro shore hardness between the individual blowing agents can be explained. It is already known from the results of the vulcanization measurement that the AZDN has the greatest influence on the torque and the associated crosslinking, which is why the lowest shore hardness is achieved here. The difference between the water/silica mixture and the microspheres can be explained by the size of the pores and the influence of the microsphere shell on the hardness. The samples foamed with water have larger pores, which can behave more elastically when the test specimen penetrates than the samples foamed with microspheres.

Another aspect of microspheres is that the microsphere shell can strengthen the matrix. When the temperature is increased, the hardness of the AZDN and water increases, which, in addition to a finer pore structure, is due to a higher final strength after extrusion as a result of faster crosslinking.

## 5. Conclusions

This study compares different blowing agents for the production of silicone elastomer foams with extrusion, focusing on the mechanical properties and the surface structure. The difference between chemical and physical blowing agents is striking since chemical blowing agents have a significant influence on the crosslinking reaction. In terms of density, the lowest density can be achieved with the chemical blowing agent, regardless of the temperature. For the physical blowing agents, water/silica mixture, and microspheres, it is noticeable that an increase in temperature leads to a further reduction in density. In the case of hardness, the lowest values can also be achieved with the chemical blowing agent, although there is an increase in hardness as a result of an increase in temperature, irrespective of the type of blowing agent. The evaluation of the surface structure of the foams produced illustrates further differences between the various blowing agents. Compared to the other blowing agents, microspheres produce the smallest and largest pores within the foam. However, the largest pores result when using the water/silica mixture.

Against the background that chemical blowing agents may have to be replaced by physical blowing agents such as microspheres or water-based agents in the European Union in the future, there is a motivation for further research into these blowing agents. On the one hand, it must be investigated whether the density can be further reduced with a targeted variation in the blowing agent content in order to achieve a density comparable to that of chemical blowing agents so that they can be used in an industrial environment. On the other hand, the influence of the carrier substance of the water on the foam morphology should also be examined in the case of water. For further investigations into the silica used, it should also be considered whether further additives may need to be added to the blowing agent mix for reliable process control. In addition to further research into the blowing agent, the interaction of the blowing agent used with the crosslinking system should also be considered, as both mechanisms can act simultaneously during the foaming process. In the case of microspheres, there are also several potential areas of research that should be investigated in more detail. On the one hand, as described above, it should be considered whether an increase in the proportion of microspheres leads to a further reduction in density. On the other hand, it should be considered whether microspheres that are larger or expand more lead to different results in terms of density. In this context, it would also be interesting to investigate under which conditions the microspheres can be used for medical applications, for example.

## 6. Patents

DE 10 2021 119 977 A1, EP000003331682B1.

## Figures and Tables

**Figure 1 polymers-16-01181-f001:**
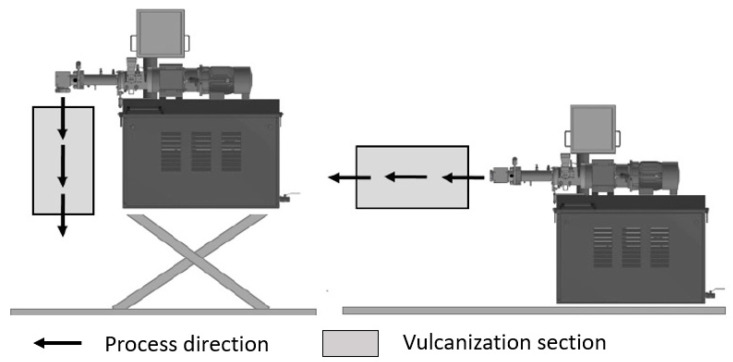
Vertical and horizontal extrusion with subsequent vulcanization based on [9].

**Figure 2 polymers-16-01181-f002:**
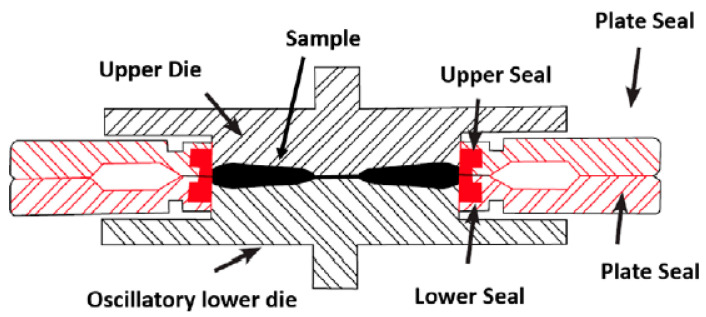
Design of an RPA Unit [4].

**Figure 3 polymers-16-01181-f003:**
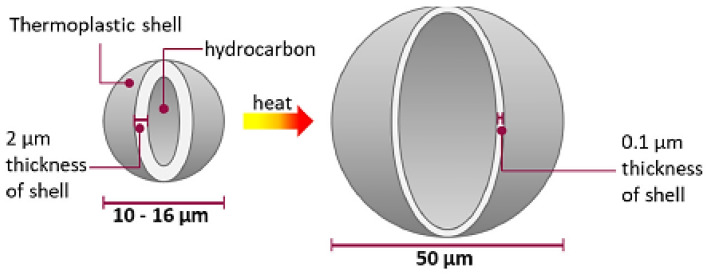
Expansion of microspheres through the supply of heat from 10–16 µm up to 50 µm (CC by S. Marl [11]).

**Figure 4 polymers-16-01181-f004:**
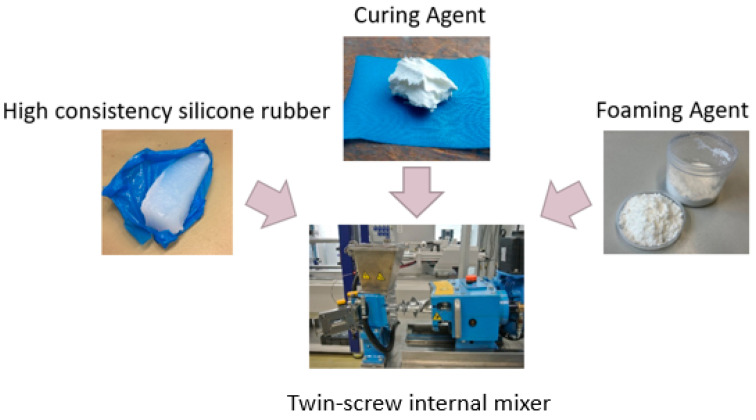
Compounding of HCR in a twin-screw internal mixer with a curing agent, foaming agent, and high-consistency silicone rubber.

**Figure 5 polymers-16-01181-f005:**
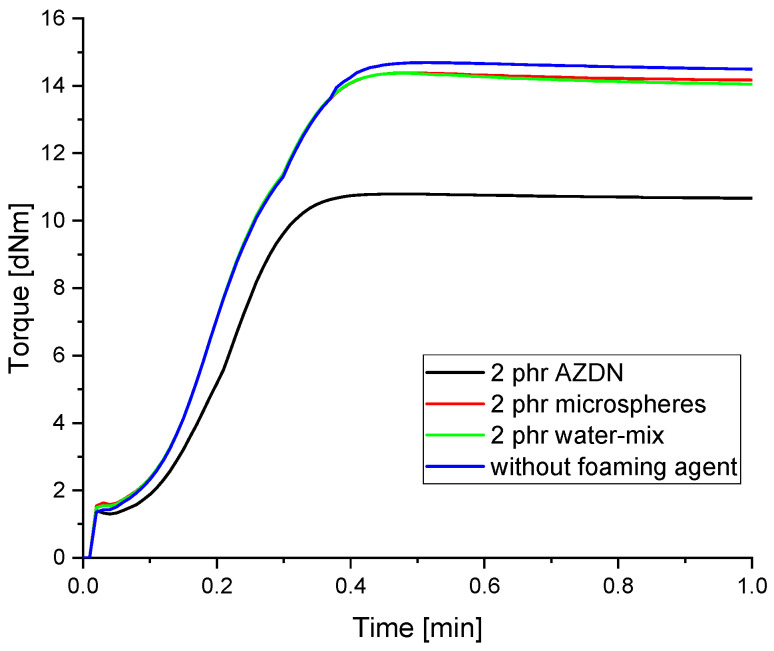
Influence of the foaming agent on the vulcanization properties of Wacker Elastosil 401-40.

**Figure 6 polymers-16-01181-f006:**
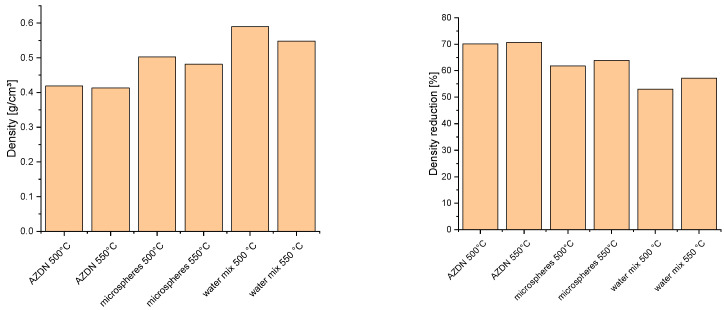
Comparison of the density and density reduction of different blowing agents.

**Figure 7 polymers-16-01181-f007:**
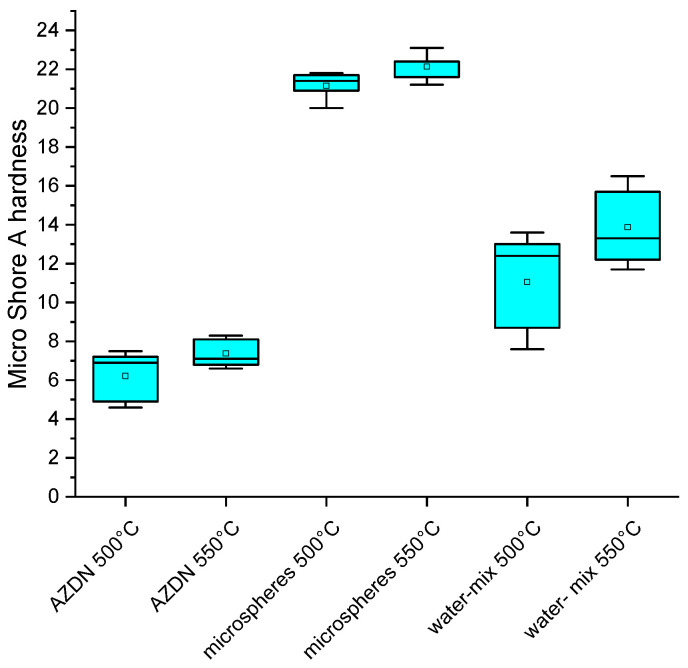
Comparison of the micro shore A hardness of different blowing agents.

**Figure 8 polymers-16-01181-f008:**
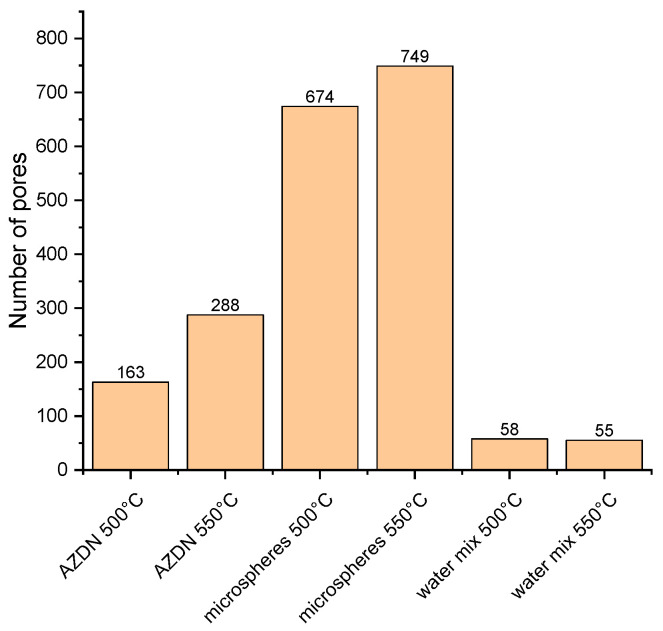
Number of pores by different blowing agents and temperatures.

**Figure 9 polymers-16-01181-f009:**
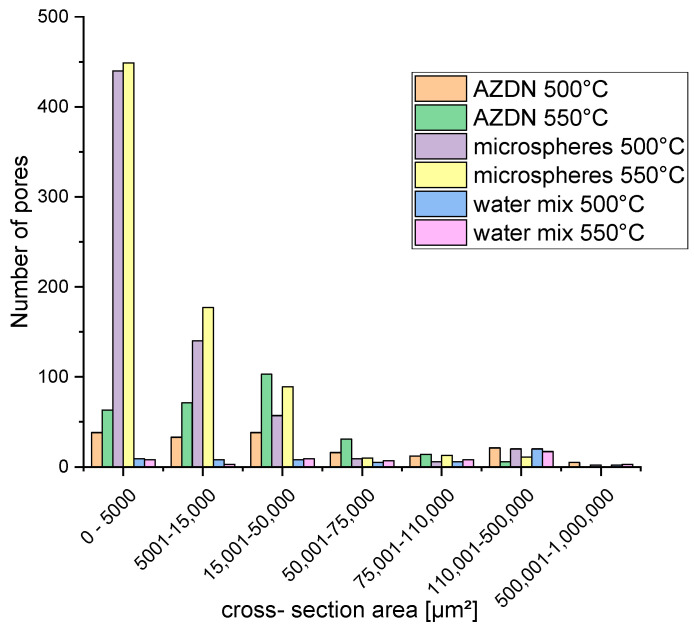
Cross-section area of the samples was determined using different blowing agents and temperatures.

**Figure 10 polymers-16-01181-f010:**
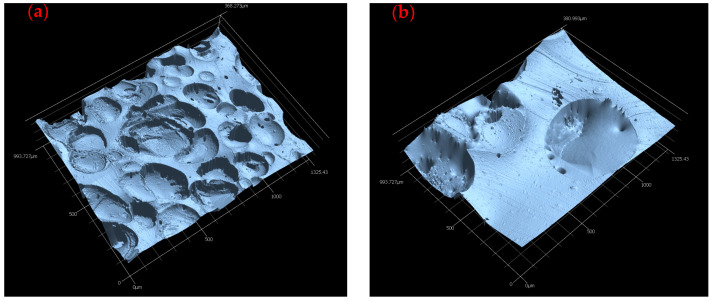
Microscopy pictures of the blowing agents AZDN (**a**) and water mix (**b**) at 550 °C were measured on the cross-section area.

**Table 1 polymers-16-01181-t001:** Process parameters for the extrusion.

Blowing Agent	Temperature [°C]	Extruder Speed [U/min]	Tool Diameter [mm]
2 phr AZDN	500	7.5	14
2 phr AZDN	550	7.5	14
2 phr microspheres	500	7.5	13
2 phr microspheres	550	7.5	13
2 phr water silica mix	500	7.5	14
2 phr water silica mix	550	7.5	14

## Data Availability

Dataset available on request from the authors.

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
