# Peer review of "High Consistency Silicone Rubber Foams"

_polymers, 2024, doi:10.3390/polym16091181_

Round 1
Reviewer 1 Report
Comments and Suggestions for Authors
The author prepared the manuscript in a novel way, however i have few queries in which the author requested to address.
1)Can you explain the observed differences in foam morphology when using different blowing agents, particularly in terms of pore size and distribution?
2)How were the reaction kinetics of the high-consistency silicone rubber affected by the use of different blowing agents?
3)In terms of practical applications, how might the findings of this study influence the selection of blowing agents for specific silicone rubber foam products?
4)How does the extrusion process contribute to the production of high-consistency silicone rubbers?
5)Are there any recommendations or future research directions suggested by the outcomes of this study, particularly in terms of optimizing the foaming process for silicone rubber materials?
Comments on the Quality of English LanguageMinor improvement required
Author Response
Dear reviewer,
thank you for reviewing my article. I have implemented your comments on the individual points as follows:
1)Can you explain the observed differences in foam morphology when using different blowing agents, particularly in terms of pore size and distribution?
I have referred here to the different foaming mechanisms and explained the parameters on which foaming depends.
2)How were the reaction kinetics of the high-consistency silicone rubber affected by the use of different blowing agents?
In the introduction, the influence of blowing agents in other rubbers was first explained in more detail, and in the discussion these findings were compared with those from this study.
3)In terms of practical applications, how might the findings of this study influence the selection of blowing agents for specific silicone rubber foam products?
In the outlook, it was pointed out that, based on these results, a future ban on chemical blowing agents could be compensated for by the water-silica mixture, for example. However, further studies are still required for this.
4)How does the extrusion process contribute to the production of high-consistency silicone rubbers?
A paragraph has been added to the introduction to explain the extrusion process. Extrusion is the most widely used process for manufacturing silicone components from HCR in terms of volume.
5)Are there any recommendations or future research directions suggested by the outcomes of this study, particularly in terms of optimizing the foaming process for silicone rubber materials?
Based on the results of this study, the outlook shows which further investigations are still necessary to make the physical blowing agents presented here interesting for industrial application. One aspect of this is the further reduction of density.
Best regards
Timo Hofmann
Reviewer 2 Report
Comments and Suggestions for Authors
-
Apart from comparing density and hardness, can we further explore the effects of different foaming agents on foam structure, heat resistance, and chemical stability to comprehensively evaluate their applicability and advantages and disadvantages?
-
Optimizing the foaming process can enhance foam quality. The dosage of foaming agents and injection time are factors worth considering. Please discuss their impact on improving the mechanical properties and surface quality of the foam.
-
There is a limited number of references, and the discussion and analysis in the paper are relatively scarce. Please increase the references to improve the quality of the paper.
-
There are multiple formatting errors in the paper, and the layout is not aesthetically pleasing. Please make the necessary modifications according to the journal's requirements.
Author Response
Dear reviewer,
thank you for reviewing my article. I have implemented your comments on the individual points as follows:
Apart from comparing density and hardness, can we further explore the effects of different foaming agents on foam structure, heat resistance, and chemical stability to comprehensively evaluate their applicability and advantages and disadvantages?
Unfortunately, as our µ- CT is currently defective, we are unable to carry out any more detailed tests in addition to those we have already performed. Unfortunately, we have no in-house facilities for testing heat and chemical resistance. If these tests are to be carried out, I would have to look for an external partner.
Optimizing the foaming process can enhance foam quality. The dosage of foaming agents and injection time are factors worth considering. Please discuss their impact on improving the mechanical properties and surface quality of the foam.
I have referred here to other sources that deal with the influence of the blowing agent content, since there are already findings in the field of liquid silicone rubber, I have included these in the consideration.
There is a limited number of references, and the discussion and analysis in the paper are relatively scarce. Please increase the references to improve the quality of the paper.
Many thanks for this advice regarding quality. I have added a few more sources to increase the quality.
There are multiple formatting errors in the paper, and the layout is not aesthetically pleasing. Please make the necessary modifications according to the journal's requirements.
Many thanks for the comment. Among other things, I have corrected the text arrangement here.
Best regards
Timo Hofmann
Round 2
Reviewer 1 Report
Comments and Suggestions for Authors
No further corrections required
Author Response
Dear reviewer,
thank you very much for your expert opinions and constructive criticism. As you had no further comments, I would like to thank you once again for your suggestions.
Yours sincerely
Timo Hofmann
Reviewer 2 Report
Comments and Suggestions for Authors
I don't see any revision marks in the manuscript.
Author Response
Dear reviewer,
thank you very much for the expert opinions and constructive criticism. I have marked the area in the newly uploaded document where I discuss the influences of the various blowing agents with the additional sources in the discussion section. This also allowed me to increase the number of sources, which you had noted. Finally, the formatting errors should also be corrected. I hope I was able to resolve the ambiguities after the first review and thank you once again for your review.
Best regards
Timo Hofmann
Round 3
Reviewer 2 Report
Comments and Suggestions for Authors
In the second revised draft, I still haven't seen any changes highlighted with any colors. I think it is your attitude towards scientific research. If you correct your attitude, I will review my opinion further.
Author Response
Dear reviewer,
please excuse my incorrect presentation of the changes. I have now highlighted the changes made in the new document in color. Based on your comments, I have tried to describe the methods more precisely and to present the results more clearly. I have also added more sources to improve the quality of the publication. I have also incorporated the influence of the amount of propellant you described. Regarding your first comment:
Apart from comparing density and hardness, can we further explore the effects of different foaming agents on foam structure, heat resistance, and chemical stability to comprehensively evaluate their applicability and advantages and disadvantages?
As our µ-CT is unfortunately still defective, we are unable to carry out any further tests on the foam structure in addition to the tests already performed. With regard to your comment that further tests should be carried out in the area of heat resistance and chemical stability, I regret to inform you that we have still not found a partner who is willing to carry this out, as we are unable to do so.
I hope I was able to resolve the ambiguities after the second review and thank you once again for your review.
Best regards
Timo Hofmann
Round 4
Reviewer 2 Report
Comments and Suggestions for Authors
Accept in present form